# A Data-Driven Review of the Genetic Factors of Pregnancy Complications

**DOI:** 10.3390/ijms21093384

**Published:** 2020-05-11

**Authors:** Yury A. Barbitoff, Alexander A. Tsarev, Elena S. Vashukova, Evgeniia M. Maksiutenko, Liudmila V. Kovalenko, Larisa D. Belotserkovtseva, Andrey S. Glotov

**Affiliations:** 1Bioinformatics Institute, 197342 St. Petersburg, Russia; barbitoff@bioinf.me (Y.A.B.); alexandretsarev@gmail.com (A.A.T.); 2Department of Genetics and Biotechnology, Saint-Petersburg State University, 199034 St. Petersburg, Russia; jmrose@yandex.ru; 3Department of Genomic Medicine, D.O.Ott Research Institute for Obstetrics, Gynaecology and Reproductology, 199034 St. Petersburg, Russia; elena.servash@gmail.com; 4Department of Biochemistry, Saint-Petersburg State University, 199034 St. Petersburg, Russia; 5St. Petersburg Branch, Vavilov Institute of General Genetics, Russian Academy of Sciences, 199034 St. Petersburg, Russia; 6Department of Pathology, Medical Institute, Surgut State University, 628416 Surgut, Russia; lvkhome@yandex.ru; 7Department of Obstetrics, Gynecology and Perinatology, Medical Institute, Surgut State University, 628416 Surgut, Russia; lbelotserkovtseva@gmail.com; 8Laboratory of Biobanking and Genomic Medicine, Saint-Petersburg State University, 199034 St. Petersburg, Russia

**Keywords:** pregnancy complications, genome-wide association study, gestational diabetes, preeclampsia, preterm birth, placental abruption, genetic variant

## Abstract

Over the recent years, many advances have been made in the research of the genetic factors of pregnancy complications. In this work, we use publicly available data repositories, such as the National Human Genome Research Institute GWAS Catalog, HuGE Navigator, and the UK Biobank genetic and phenotypic dataset to gain insights into molecular pathways and individual genes behind a set of pregnancy-related traits, including the most studied ones—preeclampsia, gestational diabetes, preterm birth, and placental abruption. Using both HuGE and GWAS Catalog data, we confirm that immune system and, in particular, T-cell related pathways are one of the most important drivers of pregnancy-related traits. Pathway analysis of the data reveals that cell adhesion and matrisome-related genes are also commonly involved in pregnancy pathologies. We also find a large role of metabolic factors that affect not only gestational diabetes, but also the other traits. These shared metabolic genes include *IGF2*, *PPARG*, and *NOS3*. We further discover that the published genetic associations are poorly replicated in the independent UK Biobank cohort. Nevertheless, we find novel genome-wide associations with pregnancy-related traits for the *FBLN7*, *STK32B*, and *ACTR3B* genes, and replicate the effects of the *KAZN* and *TLE1* genes, with the latter being the only gene identified across all data resources. Overall, our analysis highlights central molecular pathways for pregnancy-related traits, and suggests a need to use more accurate and sophisticated association analysis strategies to robustly identify genetic risk factors for pregnancy complications.

## 1. Introduction

Around 15% of all pregnant women will develop complications that could lead to maternal and fetal morbidity and mortality [1]. The most common complications of pregnancy include ectopic pregnancy, pre-eclampsia, gestational diabetes mellitus, small gestational age and preterm birth [1]. Despite numerous studies, the etiology and pathogenesis of these disorders remain unclear, and no efficient methods exist for their diagnosis, prevention, and treatment.

There is increasing evidence that, in some cases, pregnancy complications may have common pathogenetic mechanisms [2]. Pregnancy complications often share many risk factors, as well as biochemical and molecular markers [3,4,5]. Epidemiological observations indicate that pregnancy complications are risk factors for each other in one pregnancy as well as in the next ones [2,5]. Different pregnancy pathologies are often associated with similar abnormalities, such as incorrect trophoblast invasion, spiral artery transformation, defective placental development, increased secretion of inflammatory factors, oxidative stress, or endothelial dysfunction [6,7]. Despite epidemiological and pathological evidence, it is unclear whether there are common molecular pathways underlying the various pregnancy complications.

Several studies have demonstrated that there is an inherited component in most common pregnancy complications. Various strategies of genetic analysis, such as candidate gene, genome-wide association, and linkage studies, have been applied in different populations to identify genetic variants (including single nucleotide polymorphisms (SNPs)) associated with pregnancy complications [8,9,10,11,12]. Candidate gene analysis usually involves one or several pre-selected genetic variants that are tested in a small cohort of samples (usually, up to 1000) (e.g., [13,14]). Such studies can sensitively identify association between genetic changes and complex traits; however, they are prone to false positive results. In contrast to candidate gene analyses, genome-wide association studies (GWAS) allow for analysis of variants all across the genome and have high specificity (reviewed in [15]), but require much larger sample sizes compared to other approaches.

Various studies performed in the past decades identified numerous SNPs in different genetic loci to be associated with pregnancy complications [8,9,10,11,12]. Several SNPs have been found to be simultaneously associated with the risk of multiple pregnancy complications ([8,9,10,11,12]). The overlap in associated genetic variants between pregnancy complications also suggests that they may share common etiologic mechanisms. A comprehensive analysis of pregnancy complications and known genetic variants associated with their risk may be important to understanding pathogenetic processes and identification of possible common molecular pathways that lead to these conditions.

While reviewing the findings about the genetic architecture of pregnancy complications, we decided to undertake a systematic approach by retrieving data from various publicly available datasets and resources. We use literature-based gene lists, variants reported in genome-wide association databases, and an independent UK Biobank cohort [16] to analyze the genetic factors of pregnancy complications (overview of the analysis strategy is given in Figure 1). Our analysis pinpointed key molecular pathways that are important for pregnancy-related traits, and provided valuable insights into pathological mechanisms behind these traits.

## 2. Results

### 2.1. Pregnancy Complications Included in the Study

At first stage of our analysis, we obtained data for pregnancy-related traits from the three major data sources: the Public Health Genomics and Precision Health Knowledge Base (PHGKB v6.2.1) HuGE Navigator [17], the National Human Genome Research Institute (NHGRI) GWAS Catalog [18], and the UK Biobank (UKB) genetic and phenotypic dataset (summary GWAS statistics provided by the Neale lab were used). We primarily focused on the four most common pregnancy complications that require clinical decision making (gestational diabetes mellitus (GDM), placental abruption (PA), preeclampsia (PE), and preterm birth (PTB)). These four traits are the most commonly studied ones. Increasing interest in finding genetic markers associated with them is not surprising since family-based and ethnic studies confirm the possibility of a genetic contribution to the risk of all pathologies [8,9,10,11,12]. For HuGE Navigator, we manually searched for data on these major traits, For GWAS Catalog and UK Biobank resources that represent large-scale GWAS data (that often lack statistical power and are thus less sensitive) we decided to conduct a broader search, selecting pregnancy-related traits automatically using a set of keywords, with additional curation step performed afterwards. Such filtering resulted in a set of 25 and 45 traits obtained from each resource, respectively.

### 2.2. Genes Implicated in Pregnancy Complications

We began our in-depth review of the data by analyzing lists of candidate genes obtained from the HuGE Navigator resource. This resource provides a high-level view of the genetic architecture of the disease by aggregating published data on genetic associations using automatic literature search. These published associations mostly represent the results of candidate gene studies; however, some GWAS-based studies are also included in the data. Importantly, gene lists at the HuGE Navigator resource tend to contain multiple genes not actually associated with the condition under investigation due to the specific features of automated text analysis algorithms. Hence, we first curated the lists in order to get rid of irrelevant genes (see Materials and Methods for more details).

Overall, 996 genes were identified to be associated with at least one of the conditions before curation, with only 555 of these (56%) passing the manual curation step (uncurated and curated sets are available in Appendix A). After curation, 284 genes were found to be linked to PE; 233—to PTB; 170—to GDM, and only 42—to the PA. Out of the total of 555 genes identified as associated with at least one trait, a certain number of genes (139, 25%) were found to be implicated in at least two of them. Importantly, the highest number of shared genes was observed for PE and PTB (103 genes, overlap index = 0.44 (44% overlap)), which corroborates the fact that PE is one of the most common reasons for the preterm delivery. We also found 4 out of 555 genes to be linked to all four pathologies (*IGF2*, *SERPINE1*, *NOS3*, *PPARG*) (Figure 2a). The products of these genes are mostly important to pregnancy progression, and their defects might lead to placental dysfunction, which play a significant role in the pathogenesis of all the conditions studied.

We then went on to analyze the enrichment of molecular pathways for each individual trait using the Molecular Signatures Database (MSigDB) resource [19]. We have conducted the enrichment analysis using the canonical pathway list (KEGG, Reactome, Biocarta and other manually curated sets). For each of the traits, we found more than 80 pathways as significantly enriched gene sets, with the highest number of pathways enriched for PTB (336).

For PE, the top hits were pathways related to cytokine signaling (such as cytokine-cytokine receptor interaction [KEGG] and antigen processing and presentation (enrichment mainly driven by the major histocompatibility complex (MHC) locus) (Figure 2b). Another group of notable enriched pathways comprised genes related to extracellular matrix biogenesis (including core matrisome genes, secreted factors [20], and genes involved in formation of focal adhesions). Very similar groups of molecular pathways were found to be associated with PTB, with a notably strong enrichment of specific immune signaling pathways.

For GDM, we found that the most strongly enriched pathways are linked to metabolic signaling pathways, such as the *PPAR* signaling pathway, adipocytokine signaling, and *HNF3B* pathway, while many immune system-related gene sets were also identified as enriched ones (Figure 2b). Genes implicated in the PA were also enriched for the components of the *HNF3B* pathway, and, the strongest enrichments corresponded to genes involved in nuclear receptor activity.

We then analyzed how many gene sets identified as enriched for each of the 4 pathologies were shared by all 4 traits. Out of 501 canonical pathways identified as enriched for at least one trait, 264 (52%) were enriched in at least 2 gene lists, and as many as 30 gene sets were common for all 4 pathologies (Figure 2c). This result is notable as the overlap between molecular pathways was much more significant compared to the overlap between individual genes, indicating that the same processes, but not always the same genes, contribute to the studied pathologies. Shared molecular pathways can be broadly divided into two major subgroups: (i) immune-system related processes and pathways (such as interferon and interleukin signaling); and (ii) blood vessel related pathways (e.g., platelet activation and angiopoietin receptor pathway).

**Figure 2 ijms-21-03384-f002:**
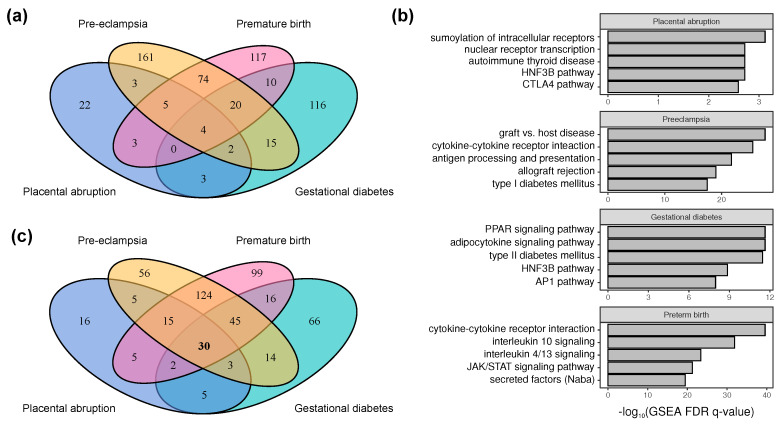
HuGE Navigator-based analysis of the genes involved in pathogenesis of pregnancy-related traits and diseases. (**a**) Venn diagram representing the number of candidate genes shared by the four major traits included in the analysis. (**b**) Top-5 enriched molecular pathways for each of the traits identified using clusterProfiler [21] analysis with MSigDB canonical pathways. (**c**) Venn diagram representing the overlap between top-100 enriched gene sets for each pathology. Note that on (**a**,**c**), preeclampsia and premature birth share the largest number of individual genes and enriched pathways.

### 2.3. An Overview of the Genome-Wide Associations for Pregnancy-Related Traits

The analysis of candidate gene lists provides an important snapshot of the genetic research in the field of pregnancy complications. However, much of the associations reported in HuGE Navigator have low statistical support and hence are not always reproducible. Genome-wide association studies, on the other hand, usually are more robust but have lower sensitivity compared to candidate gene studies. We next asked whether published GWAS studies could identify the association of genes and molecular pathways identified using HuGE Navigator-based analysis with pregnancy-related traits. To answer this question, we retrieved all published genetic associations from the National Human Genome Research Institute (NHGRI). When retrieving the variants, we used a looser *p*-value threshold ($p<10^{-5}$) in order to consider more candidate associations. To increase sensitivity, we included all pregnancy-related traits in the analysis while still focusing on 4 major ones—GDM, PA, PE, and PTB. After retrieving the SNP data, we further curated the set of variants to remove SNPs that have been withdrawn or replaced in the dbSNP variant database. We also excluded variants without any information about the genomic location and/or allelic states of the locus. Overall, we identified 201 individual variants with reported associations with 25 phenotypes (statistics shown in Figure 2a). The complete list of variants can be found in the Appendix A.

For further analysis, we grouped similar traits into several major groups to analyze their genetic architecture. These groups included (i) PE, (ii) GDM (and related glycemic traits associated with pregnancy, such as fasting glucose concentrations in pregnancy), (iii) PTB (early, moderate, and late), (iv) PA, and (v) midgestational cytokine levels.

The highest number of variants have been reported as associated with PTB (110 variants, 55% of all variants retrieved from the GWAS Catalog) (Figure 3a). For PE, we found 18 candidate associated variants; 22 variants were found to be associated with GDM and glycemic traits in pregnancy, and 9 variants—with PA. Interestingly, we found very little overlap between the GWAS Catalog lists of mapped genes and the HuGE Navigator-based lists (see below), and none of the important shared genes identified in Figure 2a were found as associated with any trait in the GWAS Catalog. Hence, we turned our attention to pathway analysis of the data.

We found 15 variants with multiple GWAS Catalog associations. These variants are listed in Appendix A. These variants correspond to several well-known target genes, such as the *GCKR* gene encoding a glucokinase regulatory protein that is known to be involved in the pathogenesis of diabetes mellitus. *GCKR* is associated with glycemic traits measured in pregnancy and is also reported as a target gene in the HuGE Navigator.

We then performed functional annotation of the obtained variants using molecular pathway analysis. As the matched reference information for the studies is not readily available, we utilized interval-based enrichment strategy implemented in the INRICH software [22]. Prior to analysis, variant coordinates were transformed into genomic regions spanning 100,000 nucleotides around the variant, and the resulting intervals were merged if overlaps were present. The resulting set of intervals was used for INRICH analysis. Unfortunately, our analysis failed to identify any molecular pathways enriched at the genome-wide associated loci at the level of FDR-adjusted p<0.05.

At the same time, we observed several notable candidate associations that were nominally associated at the level of unadjusted p<0.01. These pathways include T cell-related pathways for PE (including CTLA4, TCRA, and PD-1 pathways). The most strong candidate enrichment was observed for the genes in the IL-12 pathway [PID]. These results are concordant with the analysis of the HuGE Navigator-derived gene lists.

We found the strongest association of tight junction genes with the GDM markers (p=0.0009; adjusted *p*-value 0.19). The second strongest enrichment also corresponded to the immune system-related pathway, the T-cell receptor (TCR) pathway.

For loci associated with PTB, we found that the strongest enrichments correspond to the gene sets linked to fertilization [Reactome] and sperm motility [Reactome]. We also observed a candidate association between PTB and SKP2-E2F pathway [Biocarta], as well as a set of cell cycle-related pathways (like p53 pathway, Rb pathway, and others). For loci associated with PA, we did not find any potentially enriched molecular pathways.

### 2.4. Genome-Wide Association Studies of Pregnancy-Related Traits in the UK Biobank Cohort

Genome-wide associations usually provide robust statistical results; however, the reproducibility of the results in independent cohorts might be far from 100%. Hance, we moved on to analyze the genome-wide associations in a separate UK Biobank cohort. The UK Biobank resource provides an excellent opportunity to analyze the relationship between traits and to identify genetic factors influencing different traits. Many studies have recently been conducted using the freely available GWAS summary statistics derived from the UK Biobank cohort, with several studies focusing on global overview of the genetic architecture and pleiotropy in complex traits (e.g., [23,24]). UK Biobank features extensive phenotyping of individuals, with multiple related traits recorded for each participant. To avoid potential effects of trait misclassification, as well as to increase the sensitivity of our analysis, we retrieved summary statistics for all pregnancy- and reproduction-related GWAS datasets listed in Appendix A (45 in total).

Reconstruction of the phenotypic correlation (see Methods) of these traits identified multiple distinct clusters of traits. Expectedly, we observed that pre-eclampsia and eclampsia clustered together with the different traits related to hypertension in pregnancy (such as hypertension complicating pregnancy, childbirth, and puerperium) (Figure 3b). At the same time, we observed a very weak phenotypic correlation between the PE cluster and GDM (both ICD-10 and self-reported). This might indicate that, despite the fact that GDM is a common predisposing factor for PE, these traits have, to a certain extent, distinct genetic architecture. This finding is concordant with the HuGE Navigator-based analysis that showed low degree of overlap between target genes for these two traits (66 out of 722 genes, overlap index = 0.146 (15% overlap)). Surprisingly, we also observed low phenotypic correlation between preterm delivery and a cluster of pre-eclampsia-related traits.

After the analysis of phenotypic correlations, we went on to investigate the overall SNP-based heritability estimated from each of the 45 GWAS analyses. Out of 45 datasets considered, only 4 phenotypes showed univariate heritability estimates higher than 1% ($h^{2}>0.01$), with only one of those having statistically significant non-zero heritability (“Ever had stillbirths”). Concordantly with the results of the heritability analysis, we found very little genome-wide associations across all 45 datasets. While all of the GWAS datasets showed no signs of inflation due to under-correction of important covariates judging by the genomic inflation factor (λGC, [25]) values, we observed more than one genome-wide significant variant for only 1 trait, with quantile-quantile plots of the GWAS *p*-values showing no evidence of the association signal for all but four individual traits (including antepartum haemorrhage, APH (phenotype code O46, this trait is related to PA [26]), labour and delivery complicated by umbilical cord complications (O69), maternal care for other conditions predominantly related to pregnancy (O26), and hypertension complicating pregnancy, childbirth, and the puerperium (I9_HYPTENSPREG, correlated to PE (see above))).

All four traits showed very few variants with a genome-wide significance; hence, we decided to consider (similarly to GWAS Catalog-based analysis) as target variants all SNPs with a *p*-value $p<1\times 10^{-5}$. Such filtering resulted in 482 candidate variants potentially associated with the 4 aforementioned traits. These variants corresponded to 138 non-overlapping candidate associated genomic regions spanning 327 candidate genes (a complete list of regions can be found in Appendix A).

We then analyzed a set of loci that showed the strongest and significant (at least 1 SNP with $p<5\times 10^{-8}$) association with some of the traits (the loci are listed in Table 1). We observed a significant association between hypertension in pregnancy and the region on chromosome 2 with an index variant rs371385421 (2:113052585:C:CTGA). This variant falls into the intron of the *ZC3H6* gene and is a significant cis-eQTL for the *FBLN7* gene located upstream of the variant. *FBLN7*, in turn, encodes a protein called fibulin-7 that belongs to a large family of calcium-binding proteins, fibulins. Fibulins are components of the extracellular matrix (ECM), and are commonly viewed as bridges between different ECM proteins that stabilize the tissue structure [27]. Hence, the association between a regulatory variant for *FBLN7* and a pre-eclampsia-related trait corroborates findings about the role of extracellular matrix in pregnancy complications.

Another notable novel association corresponded to the rs59654075 (4:5427052:G:A) variant in the *STK32B* gene. This variant is associated with the umbilical cord complications, and the most likely causal gene, *STK32B*, is reported to be associated with a variety of pregnancy-unrelated traits, such as essential tremor [28]. A related gene, *STK32C*, was reported 2 times as associated with PTB in the GWAS Catalog. Finally, we found an association between antepartum haemorrhage and the rs10241971 variant that is located downstream of the *ACTR3B* gene. This gene encodes a protein that participates in the actin cytoskeleton regulation, and variation in the expression of this protein might affect pregnancy-related traits through alterations in cell adhesion and mobility.

Out of the total 482 candidate variants identified across the UK Biobank cohort, none were overlapping with the set of 201 GWAS Catalog-based variants. At the same time, we found 4 overlapping associated regions between the UKB and GWAS Catalog data, and 9 common genes located at candidate associated loci.

The most notable gene common to GWAS Catalog and UKB datasets is the *KAZN* gene that is associated with 3 out of 4 traits in the UKB data: gestational hypertension, APH, and the delivery complicated by umbilical cord complications. The rs55889542 variant in this gene is reported as associated with moderate-to-late PTB in the GWAS Catalog ([29]). This gene encodes a protein named kazrin that functions in the assembly of desmosomes ([30]). This function of the gene is concordant with a substantial role of cell-to-cell contacts in pregnancy, which is also highlighted in the gene list analysis (Figure 2).

The other common region between UKB and GWAS Catalog is located on chromosome 9 and spans the *TLE1* gene encoding a transcriptional corepressor that is involved in tumor progression, cell growth, and inflammation processes [31]. The *TLE1* locus is associated with antepartum haemorrhage in the UKB dataset and with the moderate-to-late PTB in the GWAS Catalog data. The *TLE1* gene appears to also be associated with a variety of anthropometric traits according to the GWAS Catalog. This gene is also listed as the candidate gene for GDM in the HuGE Navigator. One more shared gene is the *CCSER2* protein-coding gene that corresponds to the coiled coil serine-rich protein. This protein is viewed as a potential housekeeping gene [32] and is involved in UV response and T-cell proliferation (as recorded in MSigDB).Among the other 6 common loci, no well-characterized protein-coding genes were observed.

Despite that several common candidate associated regions were observed in the GWAS Catalog and the UKB datasets, the overall degree of correspondence between genes implicated in different pregnancy complications among the different data resources is very low, as shown in Figure 3c. The lack of correspondence between GWAS Catalog data and the UK Biobank data can be explained by several major factors. The first factor is the low sample size and low power of the UKB GWAS datasets. The second factor potentially affecting the low degree of concordance is the specificity of the UK biobank dataset and lack of additional curation of the data (for example, adjustment for reproductive age and/or trait misclassification bias).

We also conducted pathway analysis of the UKB GWAS results using INRICH. For the hypertension complicating pregnancy, a trait highly phenotypically correlated to PE, we observed a nominally significant enrichment of the tight junction genes [KEGG], a pathway that we also identified to be enriched among GDM loci using the GWAS Catalog data. This finding is noteworthy, as the overlapping genomic regions between UKB and GWAS Catalog data could not inform us about this common molecular pathway. We also identified a nominally significant enrichment of EGFR signaling pathway [Reactome].

For other phenotypes, like APH and the labour complicated by umbilical cord complications, we identified several candidate enrichments. For example, we found a candidate association between AH and the caspase activation-related genes (both acting with external stimulus and without external ligand [Reactome]), and the members of the *RUNX3-CDKN1A* transcriptional axis [Reactome]). However, in both cases no more than 2 unique genes belonging to each pathway were identified, which makes it hard to interpret these results as even nominally significant. For the O26 trait (maternal care for other conditions predominantly related to pregnancy) no enriched pathways were identified, despite the large number of candidate associated regions.

Overall, our results highlight that, despite the overall low heritability of the traits and the lack of statistical power to confidently detect genome-wide associations, certain candidate pregnancy complication genes and molecular pathways can be identified and replicated in independent cohorts. These genes, notably, are mostly related to cell cycle control and cell-to-cell contact establishment.

## 3. Discussion

### 3.1. Pregnancy Complications and Strategies for Their Genetic Analysis

In our study we assessed the genetic architecture and relevant molecular pathways for pregnancy-related traits using bioinformatic methods, primarily focusing on 4 major pregnancy pathologies—GDM, PE, PTB, and PA. Each pathology represents a serious medical and social problem [8,9,10,11,12]. PTB refers to the birth of a baby of less than 37 weeks gestational age [9,11]. The incidence of PTB is around 10%. Premature infants have risk for short and long term complications, including disabilities and restrictions in growth and development [9,11]. PTB is now thought to be a syndrome initiated by multiple mechanisms, including inflammation, stress, and other immunologically mediated processes [9]. PA, the premature separation of the placenta from the uterine wall prior to delivery of the fetus, complicates about 1% of pregnancies and is an important cause of bleeding during the pregnancy [8]. PA is mainly caused by hypertension-related problems during pregnancy [8]. PE is characterized by the onset of hypertension and proteinuria after 20 weeks of gestation. It affects 5–8% of pregnant women and is one of the predominant causes of maternal and neonatal mortality and morbidity worldwide [1,12]. PE syndrome is caused by a variety of factors including abnormal placental function, immune-system alterations, increased inflammatory activity, abnormal balance of angiogenic and antiangiogenic factors, and metabolic changes [6,7,12]. PE can cause many serious complications, including PA and PTB [12]. All three pathologies (PTB, PA and PTB) might be caused by impaired invasion of fetal trophoblasts. GDM is defined as glucose intolerance with onset or first recognition during pregnancy [1,6]. It occurs in up to 5–14% of all pregnancies [1,10]. Complications in severe cases can include fetal macrosomia, stillbirth, neonatal metabolic disturbances, and other [10]. The increasing glucose level has potential negative effects on placental function, contributing to a higher risk of PE and other pregnancy complications [6]. Numerous studies have been conducted to look for genes that influence the development of GDM, PE, PTB, and PA.

To investigate the genetic architecture of these important traits, we used two major types of data for our analysis, namely (i) candidate gene lists and (ii) genome-wide association study results. The first one (candidate genes) comes from automated literature analysis by the HuGE Navigator, and provides information about the genes that were reported to be associated with a disorder. This information is mostly obtained in analyses of one or several candidate genes in small cohorts. As such studies are highly prevalent in the field of pregnancy complication studies, the analysis of candidate gene lists is more comprehensive and provides a high-level snapshot of the major molecular pathways and biological processes involved in the pathogenesis of traits. On the other hand, candidate gene studies are prone to high false positive rates, and most of the genotype-level associations reported in literature have weak statistical support (with a large proportion of studies reporting association *p*-values between 0.01 and 0.05 (e.g., [13,14]).

Genome-wide association studies, in turn, could provide a broader understanding of the genetic risk factors of each trait because all of the possible loci are analyzed in such studies [15]. However, as a consequence of the experimental setup, large sample sizes and/or large genetic effects are required in order to robustly identify an association between a locus and a trait. As thus, traditional GWAS studies tend to have low statistical power and are prone to false negative results, especially when the sample size is not large enough (this limitation is especially relevant to pregnancy-related traits). These assumptions can explain the low number of identified candidate variants and loci in the GWAS Catalog and UKB-based analyses (Figure 3, Results Section 2.2 and Section 2.3).

As mentioned above, large sample sizes are usually needed for identification of genetic association in GWAS analyses. In studies related to pregnancy complications, study participants should also be selected and phenotyped carefully in order to avoid biases and artifacts in the analysis. This requirement hinders the use of publicly available biobank-scale datasets, such as the UKB dataset, in pregnancy-related studies. In fact, sample size limitations and improper case-control matching strategy might explain the lack of genome-wide association signal for most of the traits considered in the UKB data (see Results Section 2.3). At the same time, biobank-scale genetic datasets provide an excellent opportunity to assess the genetic and environmental correlation of traits which, for example, demonstrated lack of significant genetic correlation of GDM, PE, and PTB (Figure 3b).

Overall, the two data types considered in our analysis complement each other and allow us to dissect shared and specific components of the genetic architecture of the studied traits. While gene-level and pathway-level analysis of all candidate genes (HuGE Navigator-based) informs us on the generic processes behind the traits of interest, genome-wide association studies could point to the main loci and processes that have the greatest and most reproducible effects on the phenotype.

It is important to mention, however, that our analysis has several notable limitations. First, the conventional genome-wide association studies are conducted using only common variants. At the same time, rare variants might contribute substantially to disorders like pregnancy complications that are highly disfavored by evolution. There are several approaches for the analysis of rare variant associations (RVAS), with burden testing being the most commonly used strategy [33]. Further analysis of the genetic architecture of pregnancy complications using RVAS approaches may help in finding the missing heritability of these complex traits.

Secondly, differences in study design might substantially contribute to the overall degree of concordance between different sources. These differences include: (i) varying criteria used for diagnosis, as well as for inclusion or exclusion of samples; (ii) technical differences in data collection and analysis; and (iii) population and ethnicity of study participants.

Thirdly, data resources used in our analysis predominantly comprise information about the association between genetic variants and traits. At the same time, novel techniques are being extensively applied nowadays to investigate the molecular basis of pregnancy pathologies including, but not limited to, transcriptome and methylome analysis. One of the hottest research topics in the field of such functional genomics is the analysis of microRNA (miRNA) in both normal and complicated pregnancy. Several miRNAs are differentially expressed in the placentas and blood from women with and without PE [1,34,35,36,37,38], PTB [1], GDM [1,39], PA [40], and other pregnancy complications [1]. To date, several microRNAs have already been identified as promising biomarkers for pregnancy complications. It was shown that serum levels of miR-210 and miR-155 have high predictive value for PE (ROC/AUC of 0.90 [38]). Further comparative analysis of these data can help understand the molecular mechanisms of pregnancy complications and identify novel candidate genes.

### 3.2. The Genetic and Molecular Basis of Major Pregnancy Disorders

Our multi-perspective analysis of pregnancy-related traits allowed us to reveal major molecular pathways involved in their pathogenesis and pinpoint key genes that are central to the pathological processes. We discovered 4 genes that have been shown to affect all 4 major complications analyzed in the study (GDM, PA, PE, PTB). These genes include *NOS3*, *PPARG*, and *IGF2*.

The *NOS3* gene, encoding an endothelial nitric oxide synthase (eNOS), is one of the key genes. eNOS is an enzyme which synthesizes nitric oxide (NO) via catalyzing the conversion of L-arginine to L-citrulline. The literature data suggest that *NOS3* gene polymorphisms may be a risk factor for various pregnancy complications [41,42]. NO is a pleiotropic vasodilatator molecule that has antiatherogenic effect by inhibiting platelet aggregation, antagonizing smooth muscle cell proliferation, and suppressing leukocyte adhesion to endothelium [43]. During pregnancy NO plays a significant role in endothelial function regulation, blood pressure control, and cardiovascular homeostasis, glucose metabolism and insulin resistance [41]. The reduced NO production may predispose to pregnancy-related vascular disorders, including PE, PA, recurrent spontaneous abortion, PTB, and complications induced by GDM [42,43]. On the other hand, alterations in the NO synthesis may affect the mediation mechanisms of the inflammatory response and regulation of uterine contractility and possibly contributing to the onset of premature labor [42]. Additionally, it has shown that miR-335 suppresses the migration and invasion of trophoblast cells in PE by regulation of NOS3 [44].

The common genes associated with pregnancy complications also include *PPARG* gene, encodes the peroxisome proliferator-activated receptors gamma (PPARγ), which are ligand-activated transcription factors that regulate the expression of a number of genes involved in cell differentiation and proliferation [6,45]. Growing evidence suggests that PPARγ is necessary for the establishment and maintenance of pregnancy. PPARγ has also been shown to be involved in trophoblast differentiation, inflammatory and oxidative response, nutrient sensing and coordinating fatty acid uptake [6,45]. Changes in these pathways may lead to development of various placental-associated disorders [6,45].

Another shared gene encodes the insulin-like growth factor II (*IGF2*). *IGF2* is paternally expressed in the fetus and placenta [46]. Several studies have shown that *IGF2* is expressed in many tissues and regulates the proliferation and survival of a variety of tissues. Placental-specific *IGF2* is a major modulator of placental and fetal growth [46]. Compelling evidence obtained in recent years indicates that *IGF2* stimulates various processes that are involved in the proliferation and apoptosis of first trimester trophoblasts. This factor is also implicated in the placental supply of maternal nutrients [47,48].

In addition to notable genes influencing multiple traits, we also identified shared molecular pathways contributing to the risk of pregnancy complications. We found that cell adhesion and matrisome-related genes are commonly involved in pregnancy pathologies. The extracellular matrix (ECM) proteins play an important role in proliferation, adhesion, migration, and regulation of differentiation cells [49,50,51]. The role of ECM genes in the pathogenesis of pregnancy complications has been extensively investigated. ECM proteins are involved in almost all crucial pregnancy processes, including embryo implantation, placentation and separation of placenta after delivery and detachment [49,50,51]. Several reports suggest that changes in expression of ECM proteins are associated with development of PE, PTB and GDM [49,50,51]. Notably, different hypotheses have been made concerning the mechanism of the effects of ECM deregulation on pregnancy complications.

We identified novel genes linked to tissue structure and ECM biogenesis that also contribute to pregnancy complications. One of these loci encompasses an important ECM-related gene, *FBLN7* (Table 1) that is associated with antepartum haemorrhage (APH), a bleeding disorder that manifests from week 24 until childbirth. This trait, in turn, is linked to PA ([26]). This finding also supports a great role of ECM in pregnancy. Yet another piece of supporting evidence comes from a shared association of the *KAZN* gene with pregnancy-related traits. The protein encoded by this gene, kazrin, is a desmosome component [30], and the gene is found to be associated with multiple traits, including PTB (GWAS Catalog), hypertension complicating pregnancy, APH, and labour complications (UKB).

We identified a lot of immunity-related gene sets to be significantly enriched in the sets of implicated genes. These include broad immune system-related gene sets as well as gene sets that are related to specific immune responses (mostly T-cell related). The role of the immune system in the pathogenesis of pregnancy-related conditions is widely acknowledged (for review, see [52]). Interestingly, we also found an important gene that has not been previously paid attention to, the *TLE1* gene, that showed association with PTB in GWAS Catalog, APH -in the UKB dataset, and GDM—in the HuGE Navigator data. This might suggest that *TLE1* and the NF-κB pathway as a whole has an important role in driving diverse pregnancy complications.

Overall, data support that PE, PA, GDM, and PTB may share pathological mechanisms involving placental dysfunction. Defects of common genes are associated with irregular implantation and placentation, occurence of endothelial dysfunction, increase of oxidative stress, inflammation in placenta, which predispose to development of pregnancy complications. Interestingly, PE and PTB have the most common genes and molecular pathways. This is consistent with the current concept in the pathogenesis of these complications. PTB and PE are associated with defects in remodelling of the uterine spiral arteries in early stages of pregnancy [2]. These changes may lead to reduced perfusion in placenta, resulting in activation pro-inflammatory cytokines and oxidative stress, which lead directly to PTB as well as subsequent systemic endothelial dysfunction and PE [2].

## 4. Materials and Methods

*Public data sources*. Data for pregnancy complications were obtained from three major resources (HuGE Navigator, NHGRI GWAS Catalog, UK Biobank GWAS results (Neale lab)). For HuGE-derived gene lists, we manually curated each entry to ensure that the association between a gene and a trait was indeed reported in the original publication referenced at the PHGKB website (https://phgkb.cdc.gov/PHGKB/startPagePhenoPedia.action) or elsewhere. We considered all genes that have been shown to be directly involved in the pathogenesis of the specific trait (either through genotype-level association analysis or via any other functional genomics technology) in humans. Data from mouse models were not taken into account when curating the gene sets.

*Genome-wide association analysis*. For genome-wide association analyses performed using the UK Biobank (UKB) data we downloaded summary statistics files of 45 pregnancy-related traits provided by the Neale lab (https://docs.google.com/spreadsheets/d/1kvPoupSzsSFBNSztMzl04xMoSC3Kcx3CrjVf4yBmESU/edit?ts=5b5f17db#gid=227859291 downloaded 2019-09-22). For heritability analysis, we used summary data provided in the same data repository. Prior to all analyses, low-confidence variant sites were excluded from all datasets. The datasets were then assessed using the standard quality control methods (genomic inflation factor (λGC), inspection of quantile-quantile plots). All calculations were performed in the R programming environment. For calculation of the λGC we used the GenABEL package ([53]) For generation and subsequent inspection of quantile-quantile and Manhattan plots of *p*-values, we used the qqman [54] and CMplot (https://github.com/YinLiLin/R-CMplot) packages. The analysis of the reconstructed phenotypic correlation matrix was performed using the PhenoSpD package [55]. Absolute values of reconstructed phenotypic correlations were used for hierarchical clustering of traits. After selection of the candidate associated variants, we collapsed them into independent genomic regions. To avoid potential issues with non-matching linkage disequilibrium information, as well as to identify a broader set of potential target genes, we performed manual grouping of variants into regions of 100,000 nt long, and merged overlapping regions using the bedtools packages [56].To obtain the list of genes to be considered for further analysis, we intersected the resulting sets of non-overlapping associated regions with the GENCODE v19-derived gene intervals. All comparisons between different data sources have been conducted using custom shell scripts and bedtools.

*Pathway analysis*. For gene lists obtained from the HuGE Navigator, we performed binary gene set enrichment analysis using hypergeometric test using the clusterProfiler package [21]. The analysis was performed using the canonical pathways (cp.v.7.1) gene sets from MSigDB [19]. To perform pathway analysis of the genome-wide associations, we utilized a simple LD-aware algorithm, INRICH, that accounts for physical proximity of genes in a pathway [22]. For this analysis, variant coordinates were converted to 100,000 nt long genomic intervals, and overlapping intervals were merged using bedtools. The resulting set of non-overlapping regions was used as input to the INRICH software v.1.1. We used UK Biobank variant coordinates as the mapping file for the analysis; and MSigDB v.7.1 canonical pathways (cp.v.7.1) gene sets were used.

*Data availability*. All data and code pertinent to the analysis of the data reported here are available at https://github.com/alexandretsarev/Review_of_pregnancy_complications_variants.

## 5. Conclusions

The investigation of the genetic architecture of human complex traits is very important for finding high-efficiency diagnostic approaches, including biomarkers with high predictive power. Pregnancy complications are no exception, and understanding of the pathological processes behind these disorders is crucial for their prediction and management. All analyzed pregnancy complications are complex traits with high degree of polygenicity. Information from publicly available data resources suggests that there are several major processes that drive these disorders. These processes include immune system activity, extracellular matrix structure and cell-to-cell contacts, and metabolic signaling. We show that there are several central genes that affect all the major pathologies. These are well known *IGF2*, *PPARG*, and *NOS3* genes, as well as two genes that have not been extensively studied previously: *KAZN* and *TLE1*. We also find several novel markers of pregnancy complications that include the *FBLN7*, *STK32B*, and *ACTR3B* genes.

While major results and findings presented here can be considered as a snapshot of the current knowledge in the field, further research with well-characterized cohorts is needed to fully understand the pathophysiology and genetics of pregnancy complications. Prospective studies will show whether the identified molecular markers are independent, how important they are for each pathology separately, and whether the observed multiple associations can be attributed to the peculiar spectrum of combined diseases in the same woman. Despite having several limitations, our approach allowed us to re-evaluate the results of previous studies and narrow the search for genetic markers that control key pathways in the pathogenesis of major pregnancy disorders.

## Figures and Tables

**Figure 1 ijms-21-03384-f001:**
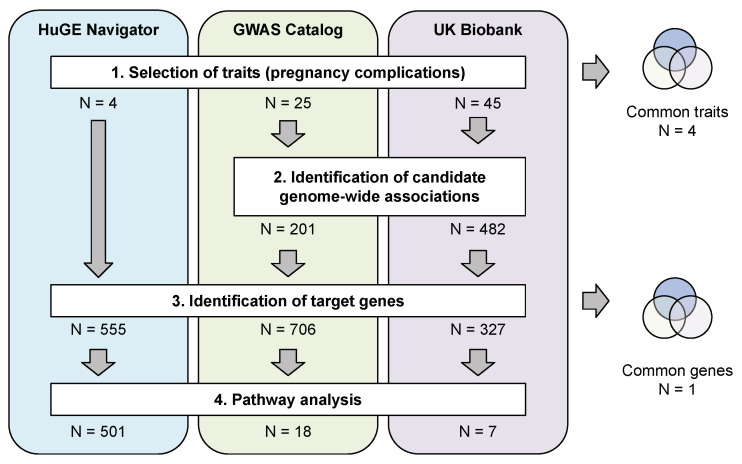
Overview of the data sources and analysis strategy. Numbers indicate the amount of variants, genes, or pathways selected at each step.

**Figure 3 ijms-21-03384-f003:**
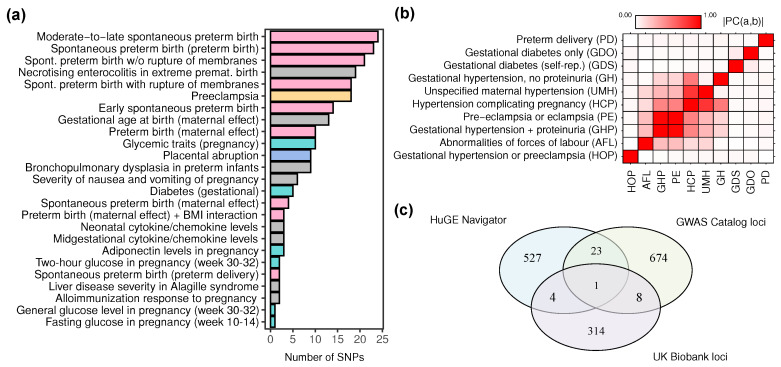
Summary of the genome-wide associations derived from the GWAS Catalog and the UK Biobank genetic dataset (**a**) Number of variants recorded in the GWAS Catalog for each of the 25 selected pregnancy-related traits. Color represents the grouping of traits (see Text), gray color represents additional traits that were not analyzed in detail. (**b**) A heatmap representing the reconstructed phenotypic correlation matrix for the selected traits in the UK Biobank genetic dataset. (**c**) Overlap between sets of genes identified from the HuGE Navigator, GWAS Catalog, and the UK Biobank dataset.

**Table 1 ijms-21-03384-t001:** Loci significantly associated with pregnancy-related traits in UK Biobank at $p<5\times 10^{-8}$.

Lead SNP Location	Lead SNP ID	Lead SNP Gene	Genes in Locus *	Trait **	*p*-Value
2:113052585	rs371385421	*ZC3H6*	*AC115115.2*, *AC115115.3*, *AC115115.4*, ***FBLN7***, *RGPD8*, *TTL*, *ZC3H6*, *ZC3H8*, *snoU13*	i9_HYP	$4.6\times 10^{-8}$
4:5427052	rs59654075	*STK32B*	*RN7SKP275*, ***STK32B***, *Y_RNA*	O69	$1.4\times 10^{-8}$
7:152604776	rs10241971	*ACTR3B*	***ACTR3B***	O46	$2.4\times 10^{-9}$
X:121644980	rs151100704	n.a.	n.a.	O26	$6.4\times 10^{-9}$

*—bold font highlights the most likely causal gene in a locus; **—UKB trait codes are given. i9_HYP. (i9_HYPTENSPREG)—hypertension complicating pregnancy, childbirth, and the puerperium; O26—maternal care for other conditions predominantly related to pregnancy; O46—antepartum haemorrhage; O69—labour and delivery complicated by umbilical cord complications.

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
