# Peer review of "A Data-Driven Review of the Genetic Factors of Pregnancy Complications"

_ijms, 2020, doi:10.3390/ijms21093384_

Round 1
Reviewer 1 Report
I was glad to review the Manuscript “A data-driven review of the genetic factors of pregnancy complications” (ijms-792309).
In my opinion, the topic is interesting and the paper is well written. Nevertheless, the authors should further improve the Manuscript:
- The Manuscript may benefit from a linguistic revision by a native English speaker in order to improve readability.
- It is essential that the Authors describe the limitations of their study. I recommend better clarifying this point.
- According to the most recent literature, epigenetic changes, in particular altered expression of selective miRNA, may play a key role in both placental-induced diseases such as pre-eclampsia and intrauterine growth restriction. It would be mandatory to discuss (at least briefly) this topic, referring to PMID: 28282763; PMID: 30256424.
- I recommend further expanding the discussion about the etiology of preterm birth, referring to PMID: 18177778.
Author Response
Reviewer #1:
I was glad to review the Manuscript “A data-driven review of the genetic factors of pregnancy complications” (ijms-792309). In my opinion, the topic is interesting and the paper is well written. Nevertheless, the authors should further improve the Manuscript:
Authors: We thank the Reviewer for positive assessment of our work as well as for the useful comments and suggestions.
The Manuscript may benefit from a linguistic revision by a native English speaker in order to improve readability.
Authors: We have rewritten several paragraphs in the Introduction and Discussion section to enhance readability. We also had our manuscript checked by a native English speaker.
It is essential that the Authors describe the limitations of their study. I recommend better clarifying this point. According to the most recent literature, epigenetic changes, in particular altered expression of selective miRNA, may play a key role in both placental-induced diseases such as pre-eclampsia and intrauterine growth restriction. It would be mandatory to discuss (at least briefly) this topic, referring to PMID: 28282763; PMID: 30256424.
Authors: We have included additional paragraphs into the Discussion to address the limitations of our approach. We also included a paragraph discussing the role of microRNAs in the traits under consideration (p. 10, lines 338 - 359).
I recommend further expanding the discussion about the etiology of preterm birth, referring to PMID: 18177778
Authors: We expanded the corresponding paragraph of the Discussion section, referring to the paper mentioned by the Reviewer (pp. 8-9, lines 282-304)
Reviewer 2 Report
In this study, the authors aimed to explore common genetic backgrounds underlying common adverse pregnancy complciations. In the beginning, the GWAS Catalog, HuGE Navigator, and pathway analysis were applied. The UK Biobank was used to validate the finding. They found out some common pathways and genetic variants underlying the adverse pregnancy outcomes.
Although it is well known that common pregnancy complications (i.e. preeclampsia, GDM, preterm labor, and placental abruption) are caused by complex genetic traits, the candidate gene association and GWAS often yield conflicting results that could not be replicated. This study tackled the dilemma by utilizing different databases. They also searched for the common pathways underlying the pathogenesis of pregnancy complications. Their findings provide valuable information to explore the pathological pregnancies and to predict pregnancy outcomes as well.
One of the major drawback of genetic association study is missing heritability. Besides of epigenetics, missing heritability may arise from rare variants. The authors need to address the limitations of their approach in the Discussion section.
Author Response
Reviewer #2:
In this study, the authors aimed to explore common genetic backgrounds underlying common adverse pregnancy complications. In the beginning, the GWAS Catalog, HuGE Navigator, and pathway analysis were applied. The UK Biobank was used to validate the finding. They found out some common pathways and genetic variants underlying the adverse pregnancy outcomes.
Although it is well known that common pregnancy complications (i.e. preeclampsia, GDM, preterm labor, and placental abruption) are caused by complex genetic traits, the candidate gene association and GWAS often yield conflicting results that could not be replicated. This study tackled the dilemma by utilizing different databases. They also searched for the common pathways underlying the pathogenesis of pregnancy complications. Their findings provide valuable information to explore the pathological pregnancies and to predict pregnancy outcomes as well.
Authors: We thank the Reviewer for high assessment of our work.
One of the major drawback of genetic association study is missing heritability. Besides of epigenetics, missing heritability may arise from rare variants. The authors need to address the limitations of their approach in the Discussion section.
Authors: We have added the discussion of this limitation to the Discussion section. The additional text reads: “It is important to mention, however, that the conventional genome-wide association studies are conducted using only common variants. At the same time, rare variants might contribute substantially to disorders like pregnancy complications that are highly disfavored by evolution. There are several approaches for the analysis of rare variant associations (RVAS), with burden testing being the most commonly used strategy (Nicolae et al., 2016). Further analysis of the genetic architecture of pregnancy complications using RVAS approaches may help in finding the missing heritability of these complex traits.” (p. 10, lines 338-344). We address other limitations of the study on p. 10, lines 338 - 359.